# Acute Pulmonary Artery Thrombosis despite Anticoagulation in Patients with COVID-19 Pneumonia: A Single-Center Retrospective Cohort Study

**DOI:** 10.3390/jcm11092633

**Published:** 2022-05-07

**Authors:** Cristian-Mihail Niculae, Ana-Maria-Jennifer Anghel, Eliza-Daniela Militaru, Laura-Georgiana Tîrlescu, Mihai Lazar, Adriana Hristea

**Affiliations:** 1Infectious Diseases Department, Faculty of Medicine, University of Medicine and Pharmacy “Carol Davila”, No. 37, Dionisie Lupu Street, Sector 2, 020021 Bucharest, Romania; eliza.manea@drd.umfcd.ro (E.-D.M.); mihai.lazar@umfcd.ro (M.L.); adriana.hristea@umfcd.ro (A.H.); 2National Institute for Infectious Diseases “Prof. Dr. Matei Bals”, No. 1, Calistrat Grozovici Street, Sector 2, 021105 Bucharest, Romania; amya.anghel@gmail.com (A.-M.-J.A.); laura-georgiana.tirlescu@rez.umfcd.ro (L.-G.T.)

**Keywords:** COVID-19, pulmonary embolism, pulmonary artery thrombosis, anticoagulation

## Abstract

(1) Background: We aimed to describe the clinical and imaging characteristics of patients diagnosed with pulmonary artery thrombosis (PAT) despite receiving anticoagulation with low-molecular-weight heparin (LMWH). (2) Methods: We retrospectively studied all hospitalized COVID-19 adult patients diagnosed with PAT between March 2020 and December 2021, who received LMWH for ≥72 h until the diagnosis of PAT. Acute PAT was confirmed by a CT pulmonary angiogram. (3) Results: We included 30 severe and critical COVID-19 patients. Median age was 62 (54–74) years, with 83.3% males, and comorbidities seen in 73.3%. PAT was diagnosed despite prophylactic (23.3%), intermediate (46.6%) or therapeutic (30%) doses of LMWH for a median time of 8 (4.7–12) days. According to their Wells score, 80% of patients had a low probability of pulmonary embolism diagnosis. PAT was localized in the lower lobes of the lungs in 76.6% of cases with 33.3% having bilateral involvement, with the distal, peripheral arteries being the most affected. At the PAT diagnosis we found a worsening of respiratory function, with seven patients progressing to mechanical ventilation (*p* = 0.006). The in-hospital mortality was 30%. (4) Conclusions: PAT should be considered in patients with severe and critical COVID-19, mainly in elderly male patients with comorbidities, irrespective of Wells score and LMWH anticoagulation.

## 1. Introduction

The acute infection (COVID-19) with the novel coronavirus, Severe Acute Respiratory Syndrome Coronavirus 2 (SARS-CoV-2) is a complex multi-systemic disease, characterized by hyperinflammation, diffuse endothelitis, platelet activation, extensive microthrombosis and angiogenesis processes [1]. 

Pulmonary embolism can develop in up to 16.5% of patients hospitalized for COVID-19 [2]. At autopsy, in more than 50% of cases, COVID-19 patients have evidence of thrombotic disease [3]. It has also been proposed that the hyperinflammatory process in the microcirculation of the lungs may cause in situ immunothrombosis, as some patients are lacking signs of deep venous thrombosis (DVT) [3,4,5]. 

Moreover, when compared with non-COVID-19 acute respiratory distress syndrome (ARDS), patients with COVID-19-associated ARDS have a significantly higher rate of pulmonary embolism diagnosis [6]. Thrombotic events can occur despite thromboprophylaxis, especially in critically ill COVID-19 patients, resulting in a five-fold increase in all-cause mortality [6]. However, there is no consensus regarding the optimal anticoagulation strategy [7,8]. Since the mechanism of thrombosis (embolism or local clot formation) cannot be systematically identified we will herein use the term pulmonary artery thrombosis (PAT), also used by other authors, covering both pulmonary embolism and in situ thrombosis [9].

The aim of this hospital-based study was to describe the clinical and imaging characteristics of patients diagnosed with PAT, despite receiving prophylactic, intermediate or therapeutic doses of low-molecular-weight heparin (LMWH).

## 2. Materials and Methods

### 2.1. Study Design and Population

This was a single-center retrospective, observational cohort study including patients admitted to a tertiary hospital with 318 beds, exclusively dedicated to COVID-19 patients, between March 2020 and December 2021. 

We included adult patients (≥18 years old) with PAT and confirmed COVID-19 diagnosed by a positive real-time polymerase chain reaction (RT-PCR) test or SARS-CoV-2 rapid antigen test, who received anticoagulation therapy with LMWH (prophylactic, intermediate or therapeutic doses) for ≥72 h until the diagnosis of PAT. Acute PAT was confirmed by a computed tomography pulmonary angiogram (CTPA).

We excluded patients with: (a) acute kidney injury or chronic kidney disease with a creatinine clearance ≤30 mL/min, and (b) confirmed or suspected DVT on admission and (c) high suspicion of PAT on admission (Wells score ≥ 2 and D-dimer ≥ 1000 ng/mL [10]) without a CTPA for exclusion before the diagnosis of PAT, and (d) chronic PAT or an episode of PAT in the last six months.

The flow diagram of the study is presented in Figure 1.

### 2.2. Definitions

Disease severity was defined as mild (normal O_2_ saturation and normal chest X-ray), medium (radiological evidence of COVID-19 pneumonia) and severe (based on at least one of the following additional criteria: peripheral oxygen saturation ≤93% in ambient air, respiratory rate > 30/minute, arterial oxygenation partial pressure to fractional inspired oxygen ratio <300 or lung infiltrates >50% of lung parenchyma) [11]. Critical disease was considered for patients on mechanical ventilation.

LMWH doses were grouped as prophylactic or therapeutic doses based on their summary of product characteristics. For patients with a BMI ≥ 40 kg/m^2^, enoxaparin as 40 mg subcutaneously twice a day and dalteparin up to 6500 units once daily were considered to be prophylactic doses [12]. Intermediate LMWH regimens were defined for enoxaparin as 40 mg subcutaneously twice a day, up to 0.5 mg/kg, according to the HEP-COVID trial [13], but we considered any dose between a prophylactic or full dose as intermediate. The LMWH regimen was prescribed upon availability in the hospital, according to the local national guidelines for COVID-19 management, suggesting the use of therapeutic doses for patients hospitalized in intensive care units (ICUs) or in those with comorbidities requiring anticoagulation.

When calculating patients’ Wells score for pulmonary embolism diagnosis, the criterion “pulmonary embolism is most likely diagnosis” was considered as “no” for all patients in the first 21 days from the disease onset. 

The PADUA prediction score for identification of patients at risk for venous thromboembolic disease (VTE) started at a baseline value of one point, given the acute infection. Reduced mobility was considered for patients with at least three days of oxygen therapy for respiratory failure. 

### 2.3. Data Collection

Demographic, clinical, laboratory and imaging data of the patients enrolled were extracted from medical and laboratory electronic charts.

### 2.4. Statistical Analysis

The descriptive statistical data for categorical variables were presented as numbers/percentages and continuous variables as medians (25–75th percentile) or means ± standard deviation (SD). We used the Shapiro–Wilk test to assess if our numerical data were normally distributed. Continuous variables were compared using the Mann–Whitney U test for data that significantly deviated from a normal distribution. A paired sample *t*-test was used for normally distributed continuous variables. Categorical variables were compared among groups using the chi-squared test. A *p* value of <0.05 was considered statistically significant. We analyzed the collected data using the Statistical Package for Social Sciences (SPSS version 23, IBM Corp., Armonk, NY, USA).

### 2.5. Ethics Statement

This study was performed in line with the principles of the Declaration of Helsinki and it was approved by the local ethics committee (C10218/2021). A waiver for patient consent was given as this was a retrospective analysis of data already on the medical records and patient data confidentiality was maintained.

## 3. Results

### 3.1. Clinical Characteristics, Anticoagulation and Imaging Data of Patients with PAT

During the study period, 30 hospitalized patients with severe and critical COVID-19 were analyzed after assessing for the eligibility criteria. Demographics, clinical characteristics and CTPA results are shown in Table 1.

Median age was 62 (54–74) years, with 83.3% males, and comorbidities were seen in 73.3% of patients, with obesity, type 2 diabetes mellitus and cardiovascular comorbidities being the most common. Considering their PADUA score, 86.6% patients were at high risk for developing VTE disease, but 80% of studied patients had a low probability of pulmonary embolism diagnosis according to their Wells score. Also, none of our patients had clinical signs of DVT at diagnosis. PAT was localized, mainly as minor PAT, in the lower lobes of the lungs in most cases, with the distal, peripheral arteries being the most affected. The left lung was more affected (36.6%) and half of the patients had at least two lobes involved, with 33.3% bilateral involvement. We found more than 50% lung involvement in 73.3% of COVID-19 patients on chest CT on admission. One patient developed a massive PAT, with hemodynamic instability, requiring thrombolysis.

PAT was diagnosed despite prophylactic, intermediate or therapeutic doses of LMWH for a median time of 8 (4.7–12) days since hospital admission. Enoxaparin was given to 14 (46.6%) patients, dalteparin to two (6.6%), nadroparin to 12 (40%) and fondaparinux to two (6.6%) patients and none of them had their level of anti-factor X_a_ monitored during LMWH treatment. 

Out of nine patients who developed PAT under therapeutic doses of LMWH, in a subgroup of four patients, based on high values of D-dimer, a first CTPA was performed, that showed the absence of PAT. In these patients, between the time points when CTPAs were performed, anticoagulation was escalated from prophylactic to therapeutic doses of LMWH (Table 2).

### 3.2. Comparative Clinical and Laboratory Data at Hospital Admission and at PAT Diagnosis 

In Table 3, clinical and laboratory data are presented comparatively between hospital admission and PAT diagnosis time points.

At the PAT diagnosis we found a worsening of respiratory function, with seven patients progressing to mechanical ventilation (*p* = 0.006). Compared to baseline, significant changes in the mean values of D-dimer (1819 vs. 7449, *p* < 0.001), C-reactive protein (124.8 vs. 40.4, *p* < 0.001), fibrinogen (613 vs. 398, *p* < 0.001) and leukocytes (8430 vs. 12,105, *p* = 0.001) were recorded at the diagnosis of PAT. No statistically significant differences were recorded between other coagulation parameters (PT, PC, aPTT) and for biological markers related to potential subclinical cardiac dysfunction secondary to PAT (CK-MB, LDH, AST, NT-proBNP).

## 4. Discussion

In this study, we described clinical, laboratory and imaging data for 30 patients diagnosed with PAT despite receiving prophylactic, intermediate or therapeutic doses of LMWH since hospital admission, for a median time of 8 (4.7–12) days. None of our patients had their level of anti-factor X_a_ monitored during their LMWH treatment.

According to their PADUA score, patients were at high risk for developing VTE as no other significant risk factors were recorded for most of them, except for obesity in eight (26.7%) patients and active malignancy in one patient. Median age was 62 (54–74) years and male gender represented 83% of patients. The majority of our patients had a low probability of pulmonary embolism diagnosis according to their Wells score. In a recent systematic review that assessed the role of five clinical scores in predicting pulmonary embolism in hospitalized COVID-19 patients, heterogeneous results on Wells scores were obtained, but five out of six studies did not report a significant association between this score and the risk of pulmonary embolism [14]. According to the authors, new prediction rules, specifically developed and validated for estimating the risk of pulmonary embolism in hospitalized COVID-19 patients, are needed.

There are very limited data on patients with acute PAT while receiving anticoagulation, mostly in patients already taking oral anticoagulation or patients from the ICU. In our study 24 patients were not in the ICU when they were diagnosed with PAT. In a case series of five patients (63–86 years, three males) with severe COVID-19 pneumonia and no other significant associated risk factors for VTE, acute pulmonary embolism was diagnosed despite adequate chronic oral anticoagulation (optimal plasma drug concentration) [15]. Similarly, Terrigno VR et al. described a case of a 38-year-old male with a recurrent pulmonary embolism in the setting of COVID-19 disease while the patient was receiving oral therapeutic anticoagulation (warfarin) for the secondary prevention of pulmonary embolism and without identified risk factors for VTE [16]. Another study, involving 184 ICU patients, found an incidence of 31% for thrombotic complications, with VTE being the most common event (27%). Out of the VTE patients, 81% developed a pulmonary embolism despite patients being on standard LMWH thromboprophylaxis [17]. Similar to our study, in this series the median age was 64 years, male gender was 76% of patients and 2.7% of them had an active cancer. In addition, Salam S et al. reported the case of a 36-year-old male with COVID-19 pneumonia diagnosed with a high-risk pulmonary embolism on intermediate dosing of LMWH for VTE prophylaxis, in which significant thrombotic risk factors were excluded [18].

In our study, therapeutic doses of LMWH were used in nine patients and PAT was initially excluded by CTPA in four of them. Considering the high D-dimer at the first CTPA and the duration of anticoagulation until the diagnosis of PAT (4–8 days), a possible explanation could be the extensive undiagnosed microthrombosis that indeed plays a role in the disease. Moreover, the hyperinflammatory process in the microcirculation of the lungs may cause in situ immunothrombosis [4,5,9,19]. This could also explain why PAT forms under anticoagulation therapy [17,20]. In their paper, Mueller et al. did not detect any thrombus in segments of the lung with normal parenchyma, supporting the hypothesis of in situ immunothrombosis [9]. Wichmann et al. described DVT at autopsy in seven of 12 patients (58%) in whom VTE disease was not suspected before death [3]. However, both pulmonary in situ thrombi due to COVID-19 inflammation and systemic embolism could be mechanisms that contribute to the full clinical scenario [9]. None of our patients had clinical signs of DVT at the time of their diagnosis of PAT, but a DVT study was not performed to look for subclinical thrombi in the lower extremities.

The localization of PAT by CTPA involved the lower lobes of the lungs in most cases, with the distal, peripheral arteries being the most affected. Similar findings were also reported also by other studies [15,17,21]. Also, most patients in our study (73%) had more than 50% of the lungs affected by SARS-CoV-2 pneumonia. These imaging data are quite similar to another paper involving patients with SARS-CoV-2 pneumonia and pulmonary embolism, as 63% of patients had more than 50% COVID-19-associated lung involvement [22]. Patients with PAT seems to have significantly higher ground glass opacity on chest CT analysis areas compared to those without PAT, especially in the lower lobes of the lungs [23]. This could be regarded as a potential pathogenetic relationship between COVID-19 pneumonia and PAT [9,23].

At the diagnosis of PAT, compared to baseline, we found a significant change in the mean values of the D-dimer, with at least a four-fold increase. Bompard et al., in a paper involving patients with COVID-19 pneumonia, described a significant difference in the median D-dimer values when comparing patients with pulmonary embolism vs. those without pulmonary embolism (9841 vs. 1285, *p* < 0.001) [22]. In their study, involving 42 COVID-19 patients, Kutsogiannis et al. found that D-dimer levels ≥5000 ng/mL had a 73% (95% CI 45–92%) sensitivity and 89% (95% CI 71–98%) specificity for predicting pulmonary embolism [6]. Moreover, in this study, D-dimer was also an independent predictor of mortality on multivariate analysis. Similar to another study, compared to baseline, inflammatory markers (C-reactive protein and fibrinogen) had lower values at pulmonary embolism diagnosis [23].

In our study, a high in-hospital mortality rate of 30% was recorded. A systematic review and meta-analysis involving COVID-19 patients with pulmonary embolism found a 45% higher mortality rate compared to general cases [24]. Also, in another study, critically ill COVID-19 patients with pulmonary embolism had a five-fold increase in all-cause mortality [20]. 

From the clinical practice perspective, PAT should be considered as a differential diagnosis in any patient with COVID-19 who has progression of respiratory failure and at least a four-fold increase in their D-dimer level, especially in weeks two to three of the disease, irrespective of their Wells score, a first negative CTPA, duration or doses of LMWH. In terms of implications for future research, for the optimization of LMWH therapy for preventing thromboembolic disease in hospitalized COVID-19 patients, monitoring the level of anti-factor Xa could be useful, especially in patients with severe or critical forms of the disease. A possible revalidation of the Wells score for pulmonary embolism for patients with severe and critical COVID-19 could also be considered.

The strengths of our study include the description of some unusual clinical presentations with a possible major impact on clinical practice in terms of differential diagnosis and anticoagulation therapy, although the small data set is an important limitation. Other limitations of this study result from its retrospective nature and the lack of data on 30- and 90-day mortality. Additionally, we were unable to account for the potential impact of underlying prothrombotic genetic risk factors or different strains of SARS-CoV-2 and there could also be subclinical PAT present, since not all severe patients were screened by CTPA. 

## Figures and Tables

**Figure 1 jcm-11-02633-f001:**
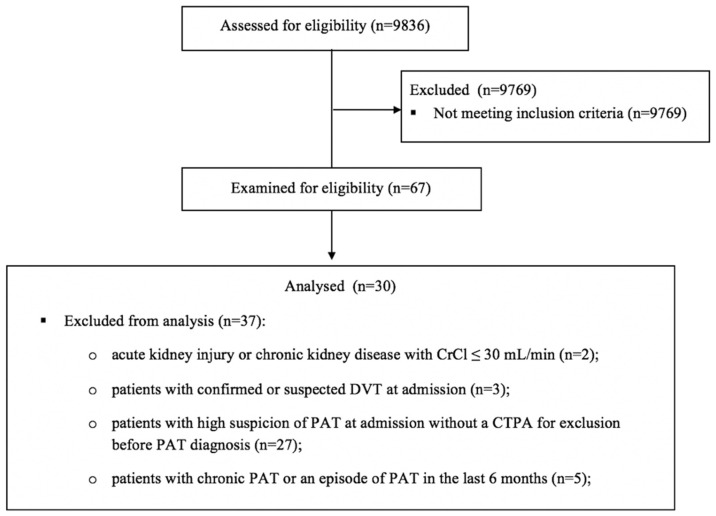
The flow diagram of the study describing the methods: participant’s recruitment, inclusions and exclusions. Abbreviations: CrCl—creatinine clearance, DVT—deep venous thrombosis, PAT—pulmonary artery thrombosis, CTPA—computed tomography pulmonary angiogram.

**Table 1 jcm-11-02633-t001:** Demographics, clinical characteristics and imaging data of patients with PAT.

Variables	All Patients (N = 30)
Male, N (%)	25 (83.3)
Age (years), median (25–75th percentile)	62 (54–74)
Comorbidities, N (%)	22 (73.3)
Obesity	8 (26.6)
Type 2 diabetes mellitus	9 (30)
Cardiovascular comorbidities,	
Hypertension	15 (50)
Congestive heart failure	2 (6.6)
Coronary artery disease	2 (6.6)
Cardiac arrhythmias	5 (16.6)
Hypertrophic cardiomyopathy	1 (3.3)
Valvular heart disease	2 (6.6)
Peripheral venous disease	1 (3.3)
Dyslipidemia	7 (23.3)
Prior stroke	1 (3.3)
Dementia	1 (3.3)
Chronic eye disorders (glaucoma)	2 (6.6)
Gastrointestinal and liver diseases	7 (23.3)
Active malignancy	1 (3.3)
Non-malignant solid tumors	3 (10)
Clinical presentation, N (%)	
Dyspnea	18 (60)
Hemoptysis	2 (6.6)
Chest tightness	3 (10)
Clinical signs of DVT at PAT diagnosis	0 (0)
Signs of right-sided heart failure	2 (6.6)
Massive PAT (hemodynamic instability)	1 (3.3)
Duration of LMWH anticoagulation until PAT (days), median (25–75th percentile)	8 (4.7–12)
LMWH anticoagulation type, N (%)	
Prophylactic doses	7 (23.3)
Intermediate doses	14 (46.6)
Therapeutic doses	9 (30)
The time from symptoms onset at PAT diagnosis, (days), median (25–75th percentile)	16 (11.7–20)
Modified two-level Wells score for pulmonary embolism, N (%)	
≤4 points (pulmonary embolism unlikely)	29 (96.6)
>4 points (pulmonary embolism likely)	1 (3.3)
Original Wells score for pulmonary embolism, N (%)	
<2 points (low probability)	24 (80)
2–6 points (intermediate probability)	6 (20)
>6 points (high probability)	0
PADUA Prediction Score at PAT diagnosis, N (%)	
1–3 points (low risk of VTE disease)	4 (13.3)
≥4 points (high risk of VTE disease)	26 (86.6)
PAT localisation by CTPA, N (%)	9 (30)
Right lung	11 (36.6)
Left lung	10 (33.3)
Bilateral	23 (76.6)
Lung base	
One lobe	15 (50)
Two lobes	10 (33.3)
3–4 lobes	5 (16.6)
Main artery	3 (10)
Lobar arteries	9 (30)
Segmental arteries	20 (66.6)
Subsegmental arteries	7 (23.3)
Chest CT analysis of COVID-19 lung involvement at PAT diagnosis, N (%)	
≤25%	1 (3.3)
25–50%	7 (23.3)
50–75%	15 (50)
≥75%	7 (23.3)
Duration of hospitalisation (days), median (25–75th percentile)	24 (20–28)
Outcome, N (%)	
Deceased	9 (30)
Discharged with no sequalae	11 (36.6)
Discharged with persistent respiratory failure	10 (33.3)

Abbreviations: PAT—pulmonary artery thrombosis, DVT—deep venous thrombosis, VTE—venous thromboembolic disease, LMWH—low-molecular-weight heparin, CTPA—computed tomography pulmonary angiogram.

**Table 2 jcm-11-02633-t002:** Duration of therapeutic LMWH, CTPA and D-dimer changes in patients with a first CTPA excluding PAT.

Cases	Duration of Therapeutic LMWH between CTPAs (Days)	D-Dimer (ng/mL) (When First CTPA Was Performed)	CTPA Changes (at PAT Diagnosis)	D-Dimer (ng/mL) (at PAT Diagnosis)
Case 1, 72-year-old man	6	11,000	Main right artery, extended to lobar, segmental and subsegmental arteries	>20,000
Case 2, 69-year-old woman	4	1930	Minor PAT, lower inferior lobe, right lung	9530
Case 3, 76-year-old man	8	14,330	Bilateral PAT—left inferior lobar artery and segmental arteries in middle lobe	2730
Case 4, 81-year-old man	8	5337	Minor PAT, lower inferior lobe, segmental arteries, left lung	8712

Abbreviations: PAT—pulmonary artery thrombosis, LMWH—low-molecular-weight heparin, CTPA—computed tomography pulmonary angiogram.

**Table 3 jcm-11-02633-t003:** Clinical and laboratory data on admission and at the time of PAT diagnosis.

Variables	At Hospital Admission (N = 30)	At PAT Diagnosis (N = 30)	*p* Value
Oxygen flow rate, N (%)			
≤15 (L/min)	12 (40)	5 (16.6)	0.006
HFOT	13 (43.3)	13 (43.3)	-
Mechanical ventilation	5 (16.6)	12 (40)	0.006
Leukocyte count (cells/mm^3^), mean ± SD	8430 ± 3520	12,105 ± 4795	0.001
Lymphocyte count (cells/mm^3^), mean ± SD	800 ± 456	959 ± 545	0.17
Neutrophils/lymphocytes ratio, median (25–75th percentile)	8.8 (6.3–14.2)	11.8 (5.2–22.8)	0.26
CRP (mg/L), mean ± SD (nv: <3)	124.8 ± 70.7	40.4 ± 41.4	<0.001
Ferritin (ng/mL), mean ± SD (nv: <290)	1689 ± 1178	1493 ± 691	0.46
D-dimer (ng/mL), mean ± SD (nv: <230)	1819 ± 3247	7449 ± 6979	<0.001
Fibrinogen (mg/dL), mean ± SD (nv: <400)	613 ± 202	398 ± 215	<0.001
PT (s), mean ± SD	13.9 ± 1.8	16.2 ± 10.7	0.26
PC (%), mean ± SD	84.5 ± 16.7	80 ± 23	0.4
aPTT (s), mean ± SD	30.2 ± 6.4	37.8 ± 23.3	0.23
CK (U/L), mean ± SD (nv: <135)	233 ± 254	137 ± 222	0.11
CK-MB (U/L), mean ± SD (nv: <16)	18 ± 14	20 ± 14	0.48
NT-proBNP (pg/mL), mean ± SD (nv: <450)	678 ± 752	2282 ± 7754	0.36
LDH (U/L), mean ± SD (nv: <246)	546 ± 234	663 ± 366	0.09
AST (U/L), mean ± SD (nv: <36)	66 ± 46	61 ± 40	0.61
ALT (U/L), mean ± SD (nv: <35)	57 ± 53	76 ± 62	0.15

Abbreviations: PAT—pulmonary artery thrombosis, HFOT—High-flow Oxygen Therapy, SD—standard deviation, nv—normal values, CRP—C-reactive protein, PT—prothrombin time, PC—prothrombin concentration, aPTT—activated partial thromboplastin time, CK—creatine kinase, CK-MB—creatine kinase-MB, NT-proBNP—N-terminal pro b-type natriuretic peptide, LDH—lactate dehydrogenase, AST—aspartate aminotransferase, ALT—alanine aminotransferase.

## Data Availability

The data presented in this study are available on request from the corresponding author.

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
