# Peer review of "Acute Pulmonary Artery Thrombosis despite Anticoagulation in Patients with COVID-19 Pneumonia: A Single-Center Retrospective Cohort Study"

_jcm, 2022, doi:10.3390/jcm11092633_

Round 1
Reviewer 1 Report
Page 8 line 194
This is a very clinically important study given optimal anticoagulation for covid 19 hospitalised patients is not known. In this study , authors have demonstrated Pulmonary thrombosis(Thromboembolism?) diagnosis despite therapeutic anticoagulation , at least in 4 out of 9 patients very clearly . This data is very useful for research
- Authors need to explain clinician discretion with regards to choosing dose of anticoagulation(not the actual LMWH product) given it is a single center study. Was there any institutional policy on heparin use?
2. Limitations need to be elaborated - especially small data set. There could be subclinical PE given not all patients were screened. LMWH activity monitoring was not done.
3. was DVT study performed to look for thrombi in lower extremities after PE was diagnosed ?
4. Whether the authors are suggesting 4 out of 9 patients clearly had pulmonary in situ thrombi due to COVID-19 inflammation and not PE which is why therapeutic anticoagulation did not work? If so, can they be rightly classified under pulmonarythrombo embolism?
5. Discussion section- should elaborate on pulmonary in situ immunohrombi
Reviewer 2 Report
The authors restrospectively studied the charateristics of acute pulmonary embolism despite anticoagulation in patients with COVID-19 pneumonia. No longer follow-up data and no different SARS-CoV-2 strains's analyses are the major limitation in this study as the authors described in this manuscript.
